# Advancing 7T perfusion imaging by pulsed arterial spin labeling: Using a parallel transmit coil for enhanced labeling robustness and temporal SNR

Ícaro Agenor Ferreira Oliveira[1,2,3]*, Robin Schnabel[1], Matthias J. P. van Osch[4], Wietske van der Zwaag[1,2©], Lydiane Hirschler[4©]

1 Spinoza Centre for Neuroimaging, Netherlands Academy for Arts and Sciences, Amsterdam, Netherlands, 2 Computational Cognitive Neuroscience and Neuroimaging, Netherlands Institute for Neuroscience, Amsterdam, Netherlands, 3 Krembil Brain Institute, University Health Network, Toronto, Ontario, Canada, 4 C.J. Gorter MRI Center, Department of Radiology, Leiden University Medical Center, Leiden, Netherlands

© These authors contributed equally to this work.
* icaro.oliveira2@uhn.ca, icarooliveira@yahoo.com.br

**Data Availability Statement:** Due to legal and ethical constraints surrounding potentially identifying information, MRI data cannot be shared

## Abstract

Non-invasive perfusion imaging by Arterial spin labeling (ASL) can be advantageous at Ultra-high field (UHF) MRI, since the image SNR and the T1 relaxation time both increase with the static field. However, ASL implementation, especially at 7T, is not trivial. Especially for ASL, UHF MRI comes with many challenges, mainly due to $B_1^+$ inhomogeneities. This study aimed to investigate the effects of different transmit coil configurations on perfusion-weighted imaging at 7T using a flow-sensitive alternating inversion recovery (FAIR) technique with time-resolved frequency offset corrected inversion (TR-FOCI) pulses for labeling and background suppression. We conducted a performance comparison between a parallel transmit (pTx) system equipped with 32 receive (Rx) and 8 transmit (Tx) channels and a standard setup with 32Rx and 2Tx channels. Our findings demonstrate that the pTx system, characterized by a more homogeneous $B_1$ transmit field, resulted in a significantly higher contrast-to-noise ratio, temporal signal-to-noise ratio, and lower coefficient of variance (CoV) than the standard 2Tx setup. Additionally, both setups demonstrated comparable capabilities for functional mapping of the hand region in the motor cortex, achieving reliable results within a short acquisition time of approximately 5 minutes.

## Introduction

Arterial spin labeling (ASL) is a versatile magnetic resonance imaging (MRI) technique for measuring cerebral perfusion as well as 4D angiographic information. ASL utilizes the water in arterial blood as a diffusive endogenous contrast agent, offering a completely non-invasive measurement of cerebral blood flow (CBF). ASL can also provide valuable insights into brain function as a more physiological alternative to blood oxygenation level-dependent (BOLD)

publicly. MRI data falls under the category of personal data under the General Data Protection Regulation (GDPR) and is subject to the policies outlined by the Royal Netherlands Academy of Arts and Sciences (KNAW). Requests for access to the data will be evaluated on a case-by-case basis, subject to a data usage agreement. For inquiries regarding data access, please contact Tomas Knapen at t.knapen@spinozacentre.nl. The analysis code is available and can be accessed via the following link: https://github.com/icaroafoliveira/ASL_7T_pTx_scripts.

**Funding:** This work was supported by a Royal Netherlands Academy for Arts and Sciences (KNAW) research grant NWO (Nederlandse Organisatie voor Wetenschappelijk Onderzoek), Number: TTW VI.Vidi.198.016 to W.Z. The funders had no role in study design, data collection and analysis, decision to publish, or preparation of the manuscript.

**Competing interests:** The authors have declared that no competing interests exist.

fMRI [1, 2]. Additionally, it presents itself as a quantitative method for measuring CBF, which is superior in temporal and spatial resolutions when compared to other methods, such as single photon emission computed tomography (SPECT) and positron emission tomography (PET) [3]. It has been extensively validated against methods that use exogenous contrast agents, such as $^{15}$O-PET [4, 5]. It is worth pointing out that the present study is primarily focused on spatial selective techniques (and then especially pulsed ASL). The advantages and disadvantages of pTx might be different for velocity-selective labeling techniques, for specific information about velocity-selective labeling (VSASL) techniques, the reader is referred to [6].

The fundamental concept of signal generation in spatially selective ASL is the manipulation of the longitudinal magnetization of water present in arterial blood to differentiate it from the magnetization of static tissue. As intravascular spins reach the capillaries and undergo exchange with brain tissue across the blood-brain barrier, the tissue magnetization is altered, allowing the acquisition of images whose contrast is proportional to cerebral perfusion after subtraction of the labeled image from a control acquisition [1].

While ASL can act as a surrogate for CBF mapping and has potential as an alternative fMRI contrast, it is important to acknowledge the challenges and limitations that ASL encounters. ASL suffers from an intrinsically low signal-to-noise ratio (SNR) due to the $T_1$ relaxation, limited temporal resolution because both tag and control images need to be acquired (two times ~3s), and limited spatial resolution and/or coverage [7]. The SNR is inherently low because the signal from the labeled spins reaching the brain tissue is only a small fraction (0.5% to 1.5%) of the static tissue signal [1, 3]. One typical solution to improve the SNR of ASL is the use of background suppression pulses to reduce the static tissue signal, thereby improving sensitivity and reproducibility [8]. Another common strategy is to acquire multiple repetitions (or multiple segments of a 3D readout) to ensure sufficient SNR [2, 9].

Scanning at a higher field strength might hold the key to improving the intrinsic low SNR of ASL as recent research has revealed that a higher static magnetic field ($B_0$) leads to a supra-linear improvement in SNR of a traditional MRI scan [10], which would also enhance the sensitivity and precision of ASL measurements. In addition to the image SNR that increases with the static field, Ultra-high field (UHF, 7T) ASL will profit from longer $T_1$ relaxation time allowing for longer inversion times (TI, for Pulsed ASL) or longer post-labeling delays (PLD, for pseudo-continuous ASL), which amplifies the perfusion-related signal and would also allow proper CBF-measurements in subjects with slow blood flow (long arterial transit times). Moreover, it would also allow longer readout acquisitions, increased brain coverage, as well as higher spatial resolution perfusion images [7, 11–14].

However, UHF ASL implementations are not trivial and also come with severe challenges such as $B_0$ and $B_1$ inhomogeneities that must be carefully addressed [15]. For example, several adiabatic inversion pulses have been specifically proposed for Pulsed ASL (PASL) to mitigate the $B_1^+$ inhomogeneity [16, 17]. In addition, the labeling efficiency can also be compromised by limited coil coverage; a typical solution for PASL sequences to cope with both challenges is the use of dielectric pads to improve the inversion efficiency and to obtain more homogeneous excitation as a secondary advantage [11, 18]. For pseudo-continuous ASL (pCASL), the labeling plane can be raised from the neck to the bottom of or even within the cerebrum to take advantage of the locally higher $B_1^+$ [19, 20].

Another possible solution for a more homogeneous transmit field is parallel transmission (pTx) technology; pTx capabilities rely on using the additional degree of freedom of the multiple channels to better control the $B_1^+$ field; for more details, please check [21]. Previous studies have optimized and evaluated strategies such as $B_1$-shimming or tailored RF pulses in ASL [22–24]. These investigations primarily aimed to enhance the labeling efficiency of pseudo-

continuous ASL (pCASL) using pTx techniques. However, it is worth noting that, for functional experiments, less SAR-intensive PASL sequences are often preferred at UHF.

In the present study, we extended the use of the pTx coil (32Rx8Tx) in a near-quadrature mode to a Pulsed ASL (flow-sensitive alternating inversion recovery, FAIR) sequence in combination with a time-resampling frequency-offset-corrected-inversion (TR-FOCI) pulse and compared the results to data obtained with the 32Rx2Tx coil on the standard 2-channel system using the same sequence and RF pulses. We assessed the performance of both coils through two experiments. The first experiment consisted of a variation of the inversion slab thickness to test the behavior of the label created by both transmit coils. Next, we used finger-tapping stimuli to investigate if the coils would yield similar fMRI results.

## Material and methods

### Participants

Seven healthy volunteers (four females, aged 25–44 years) were recruited to participate in this study. The participants enrolled between 04-11-2022 and 20-01-2023, and each participant underwent two MRI sessions on the same day, utilizing the pTx setup (32Rx8Tx) and the standard 2-channel transmit coil. The local ethics committee approved the study protocol (Amsterdam University Medical Center—location AMC), and all volunteers provided written consent after receiving comprehensive information about the experimental procedures.

### Data acquisition

Data were acquired on a 7T MRI scanner (Philips Healthcare, Best, The Netherlands) at the Spinoza Centre for Neuroimaging (Amsterdam UMC, Amsterdam, The Netherlands), equipped with a gradient set with a maximum gradient strength of 40 mT/m and 200 mT/m/s of slew rate. We utilized two commercially available head coils in our study (Nova Medical Inc, Wilmington, USA). The first coil, known as 32Rx2Tx or 2Tx, is equipped with 32 receive phased array elements and two transmit channels. This coil features a circularly polarized transmit coil design with a helmet-shaped insert containing 32 receive elements. The second coil, referred to as pTx, incorporates eight transmit elements arranged cylindrically around the head, the 32-receive insert is organized as for 2Tx receive. The pTx coil employs a close-to-circularly polarized mode achieved through B1-shimming across the entire brain using data from a separate group of volunteers [20]. No personalized RF-shim or other pTx pulses were used here to limit the comparison to the hardware used.

A FAIR Arterial Spin Labeling sequence was implemented using TR-FOCI RF-pulses [25] with 13ms of pulse duration, nominal flip angle of 2576 degrees, and B1 = 15μT. We did not optimize the adiabatic inversion pulse, and all RF attributes, including nominal flip angle and $B_1^+$ amplitude, for each setup separately, i.e., they were kept consistent across both coils. Pre- and post-saturation modules were included before and after labeling to saturate the imaging volume. We employed two non-selective TR-FOCI RF-pulse for background suppression at 1200 and 1830 ms. The readout consisted of a multi-slice, single-shot 2D EPI with FOV = 210 × 210 mm$^2$, matrix = 72 × 72, with a nominal resolution of 3 × 3 mm$^2$, thirteen ascending 3-mm slices (no gap with an inter-slice duration of 65 ms), TI/TE/TR = 2000/8.9/5500 ms, and SENSE factor = 3. Additionally, we acquired MPRAGE data with the same spatial resolution (0.8mm) and sequence parameters as in [26] during the 2Tx session.

We used two different experiments to investigate the performance of the coils. In the first experiment, the ASL FAIR sequence was acquired with a varied slab inversion thickness to study whether the effective labeling slab width is larger with the pTx setup. We used 6cm, 9cm, and 12cm larger than the volume of interest for both coil setups to vary the size of the tag

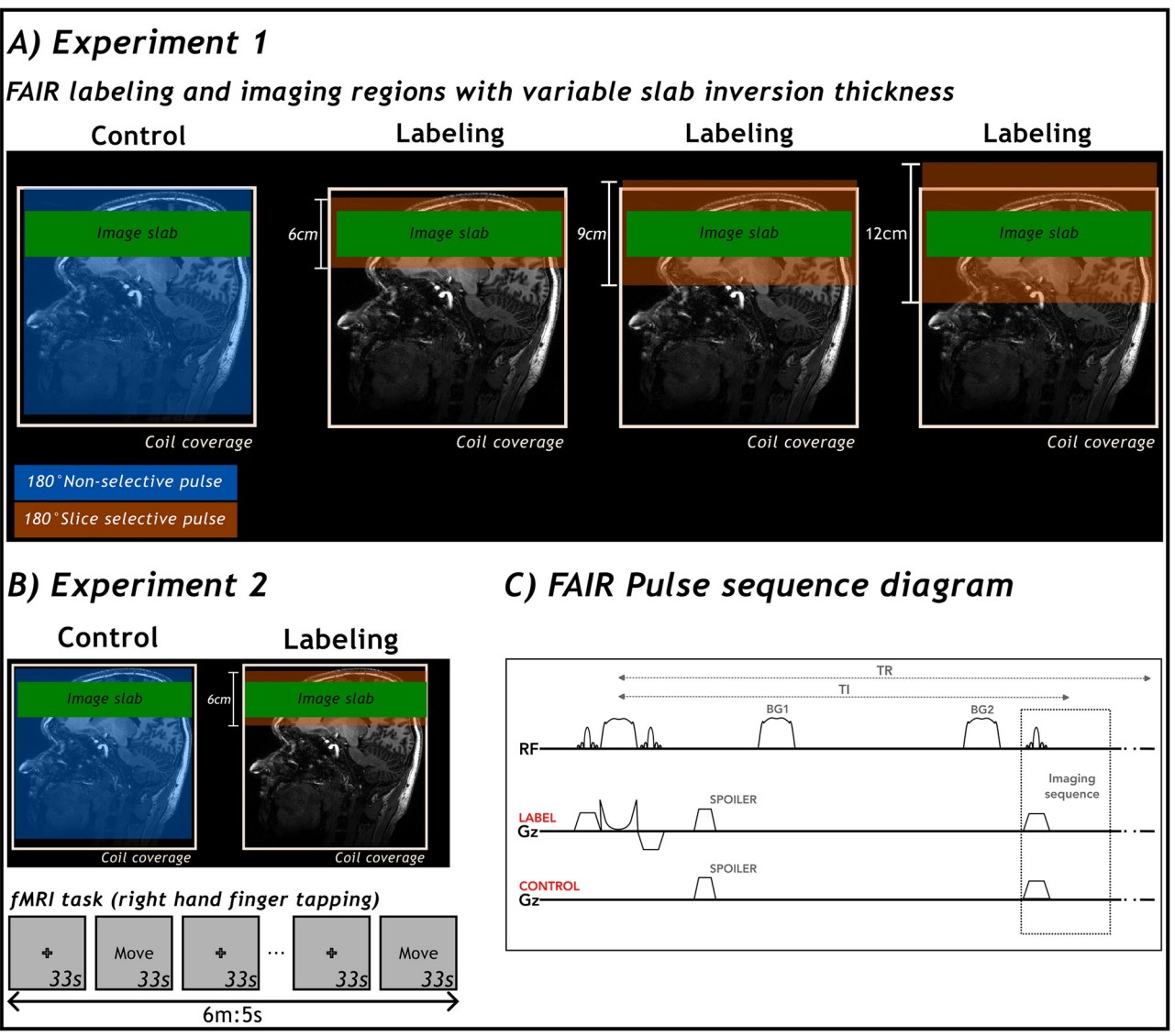

**Fig 1. Experimental design illustrating the Pulsed Arterial Spin Labeling (PASL) FAIR sequence.** In panel (A) Experiment 1, the location of the control and labeling planes is depicted, followed by an example of possible additional slab inversion thicknesses. The imaging plane is represented in green, the non-selective inversion pulse (control) in blue, the slab selective inversion pulse (labeling) in orange, and the coil coverage in red. In panel (B) Experiment 2, the protocol for the fMRI experiment is outlined along with the designed fMRI stimulus. The stimulus involved right-hand finger tapping every 33 seconds after the baseline (also 33 seconds), followed by a 33s rest condition (indicated by a cross sign). The total experiment time was 6 minutes and 5 seconds. Panel (C) shows the pulse sequence diagram with all relevant elements.

(smaller for larger slab inversion thickness), please see Fig 1A for more details. For this experiment, we acquired 12 pairs of volumes of ASL data for each slab inversion thickness.

For the second experiment, we acquired ASL-fMRI data. For this part of the study, we obtained data only from a subset of participants (P04-P07) using the protocol with the inversion slab 6cm larger than the volume of interest (Fig 1). A 33 volume-pair time course was acquired during a finger-tapping functional task on the 2-channel and 8-channel transmit systems. The right-hand finger-tapping fMRI experiment consisted of a block design alternating 33s on/off blocks for 6 minutes.

For the pTx system, a maximum $B_1^+$ value of 18μT was used for the excitation pulses; this is the default maximum for this system. For the 2Tx, a maximum $B1^+$ of 12μT was used for the excitation pulses to accommodate amplifier limitations. For the system used in the present study, changes in the B1 maximum are applied only to the excitation pulses. The inversion pulse attributes, such as B1 amplitude, pulse length, and flip angle were kept as predefined. The inversion pulses do not exceed safety limits. Apart from the excitation $B_1^+$ value, all sequence parameters were kept the same for all participants. S1 Fig in S1 File to see $B_1^+$ maps.

## Data analysis

For the first experiment, we first performed motion correction for both control and label images separately using SPM12-based (SPM12, version 7771) ASL toolbox functions available in ASLtbx [27]. The alignment between label and control as well as across the three inversion slab thicknesses (6cm, 9cm, and 12 cm) was conducted as part of the motion correction for the 2Tx and pTx data separately.

ASL image registration between coils was performed using a combination of iTksnap and the FLIRT registration software from FSL. First, the affine matrix was generated in iTksnap (version 4.0.2) [28] either manually or by using the iTk automatic registration function. The saved affine matrix was used as the initial matrix to register the 2Tx data into the pTx space using FLIRT (FSL version 6.0.6.4). To create the gray-matter and white-matter masks, we used the segmentation function from SPM12 on the averaged perfusion-weighted images. Gray and white matter binary masks for the left and right hemispheres were created manually using iTK-SNAP by editing the segmented gray-matter and white-matter tissues.

For the second experiment, motion correction was performed using the ASLtbx, followed by the use of an isotropic Gaussian kernel of 5mm in SPM12 to increase the functional signal-to-noise ratio (SNR) in the perfusion fMRI experiment. Following the preprocessing steps, the perfusion images, statistical analysis, and figure plots were generated by in-house scripts using Julia Language [29] v.1.7.2 with the following packages MriResearchTools.jl v.0.7.0, Statistics.jl v.1.8.0, DataFrames.jl v.0.20.2, GLM.jl v.1.1.1 and CSV.jl v0.8.5 and R v.4.2.1 [30] with ggplot2 v.3.4.2 and ggpubr v.0.4.0 packages [31, 32]. All in-house scripts developed for this study can be accessed through the following link (https://github.com/icaroafoliveira/ASL_7T_pTx_scripts). We used the GLM analysis as implemented in FSL (FEAT, v.6.0.6.4) for the fMRI experiment.

FAIR perfusion-weighted maps were generated through pairwise subtraction of the control and label images ($\Delta M = S_{control} - S_{label}$). It is worth mentioning that while we acknowledge the importance of robust quantification in certain contexts, in the scope of this study, the omission of CBF quantification allows for a more direct comparison of coil performance.

To investigate the performance of both coils, we first did a visual inspection of the image quality of the images produced by both coils. Quantitative measurements were based on the stability of the signal over time by calculating the coefficient of variance (CoV) and temporal signal-to-noise ratios (tSNR) of the perfusion images. The CoV is generated by calculating the standard deviation of the perfusion images over time divided by the mean of the perfusion images over time, as shown by equation [1]. The tSNR maps were generated by calculating the mean of the perfusion images over time divided by the standard deviation over time, as shown by equation [2]. Additionally, signal-to-noise ratio (SNR) and contrast-to-noise ratio (CNR) were also assessed. The SNR was calculated using equation [3], and it was defined as the mean divided by the standard deviation of the gray-matter mask. The CNR was calculated using equation [4], $\mu_{GM}$ and $\mu_{WM}$ are the mean signal intensities of gray matter and white matter regions of interest, respectively. $\sigma_{GM}$ and $\sigma_{WM}$ are the standard deviation of the signal

intensities in the gray matter and white matter regions of interest, respectively. The SNR equation using the same ROI is based on the recommendation [33] to minimize the influence of differences in the spatial distribution of the noise in parallel imaging acquisition. Both SNR and CNR equations are derived from the publicly available quality assurance pipeline (MRIQC) [34]. The statistical assessment was conducted in R version 4.2.1 [30], using Paired t-tests with alternative hypotheses that the means (pTx vs 2Tx) are different. To address multiple comparisons, we applied Bonferroni correction.

$$CoV = \frac{SD(Perfusion)}{mean(Perfusion)} \tag{1}$$

$$tSNR = \frac{mean(Perfusion)}{SD(Perfusion)} \tag{2}$$

$$SNR = {}^{\mu_{GM}}/_{\sigma_{GM}} \tag{3}$$

$$CNR = \frac{abs(\mu_{GM} - \mu_{WM})}{\sqrt{\left(\sigma_{GM}^2 + \sigma_{WM}^2\right)}} \tag{4}$$

We ran linear regression to evaluate the relationship between the perfusion signal (in terms of CoV) and the variation in the labeling slab thickness. We calculated the slope and intercept of the CoV as a function of the slab inversion thickness. Using only the 6cm slab protocol, we used a t-test to check whether there were differences in the CoV between both coils.

We also compared the tSNR values and mean and max z-scores from the motor area gray matter mask for the fMRI experiment. The tSNR values were extracted from the whole brain gray-matter mask using the fMRI dataset, we did not exclude the activated voxels in the analysis. The max and mean z-values were extracted from the Z-statistical activation maps after thresholding for z > 3.1 and p = 0.05.

## Results

Fig 2 shows perfusion-weighted images from an example participant, showing data acquired with three different thicknesses of the slab-selective inversion pulse and the 2Tx and the pTx

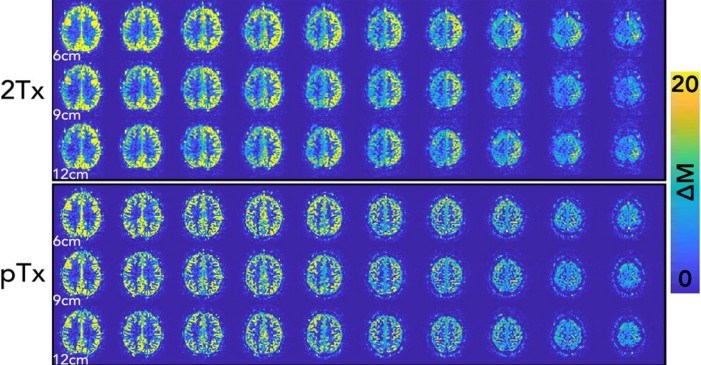

**Fig 2. Arterial spin labeling difference image (ΔM) from a single participant (P06), highlighting differences between the 2Tx and pTx systems.** The pTx system generally shows more homogeneous signals (e.g. fewer right-left differences) than the 2Tx. The increased thickness of the slab-selective inversion (6cm, 9cm, and 12cm) resulted in an intensity reduction for both the 2Tx and pTx systems.

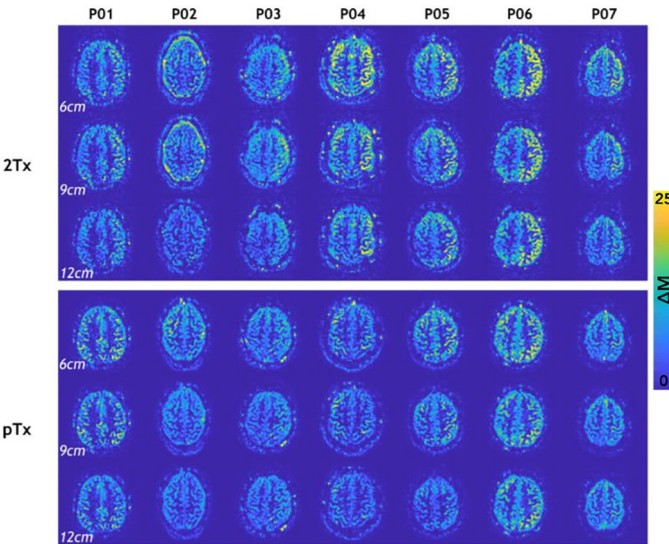

**Fig 3. Single slice (#6) arterial spin labeling difference (ΔM) from all 7 participants.** An asymmetric right-left signal pattern can be seen on the 2Tx images for all slab thicknesses and participants. A more homogenous pattern is consistently observed in the pTx images. The contrast window is the same for all perfusion-weighted images.

systems. Increasing the thickness of the slab-selective inversion resulted in an intensity reduction in the perfusion-weighted images in both coils, as expected from the effectively smaller labeling region. In addition, we noted in this participant a lower signal for the 2Tx system compared to the pTx. These results were consistent across participants (Fig 3 & S2 Fig in S1 File). Fig 3 shows example slices from the perfusion-weighted images from all participants for both the pTx and the 2Tx systems. On the 2Tx system, we consistently observed an asymmetry in perfusion signal brightness between the right and left hemispheres, whereas the pTx images do not show the same pattern.

In Fig 4, the results of the linear regression are shown. This was used to assess the relationship between the slab inversion thickness and the coefficient of variance (CoV) of the perfusion signal for each ROI and coil. Average slopes and intercepts across participants and ROIs were as follows: For the 2Tx the Right-GM ROI, slope = 0.074, intercept = 0.349, and Left-GM ROI, slope = 0.053, intercept = 0.347. For the pTx, the Right-GM ROI, slope = 0.037, intercept = 0.266; for the Left-GM ROI, slope = 0.042, intercept = 0.224. For both ROIs, the average slope across participants is slightly higher for the 2Tx compared to the pTx coil, meaning that there is more perfusion signal loss at larger inversion slab widths. The detailed output of the linear regression with the respective error and goodness of fit is provided in the supplementary material (S1-S4 Tables in S1 File). We presented a boxplot depicting the perfusion CoV for both gray-matter hemispheres and three different labeling thicknesses, emphasizing the distinction between 2Tx and pTx. A paired t-test (Bonferroni corrected) revealed a significant difference in both hemispheres for the two labeling thicknesses, 6cm, and 9cm. Additionally, a significant difference was observed for the labeling thickness of 12 cm, but exclusively in the right hemisphere. These results are visually represented in Fig 5.

To assess the level of noise between the two coils, we investigated the SNR and CNR. For the SNR the left and the right gray matter ROIs were assessed. As can be seen in Fig 6 panel A, the median SNR was higher for pTx compared to the standard 2Tx coil in both ROIs and across the different slab inversion thicknesses. The SNR distribution across participants seems

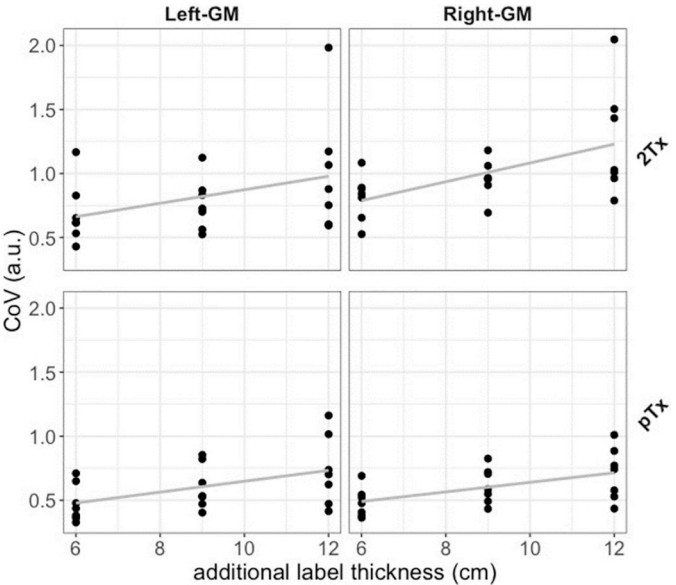

**Fig 4. Coefficient of variance (CoV) vs. additional (label thickness) slab inversion thickness for right and left masks and coils (2Tx and pTx), showing an increased signal for 2Tx than pTx.** CoV reflects the amount of noise in the perfusion-weighted images. The increase in the thickness of the slab inversion resulted in a slightly higher signal loss for the 2Tx than pTx.

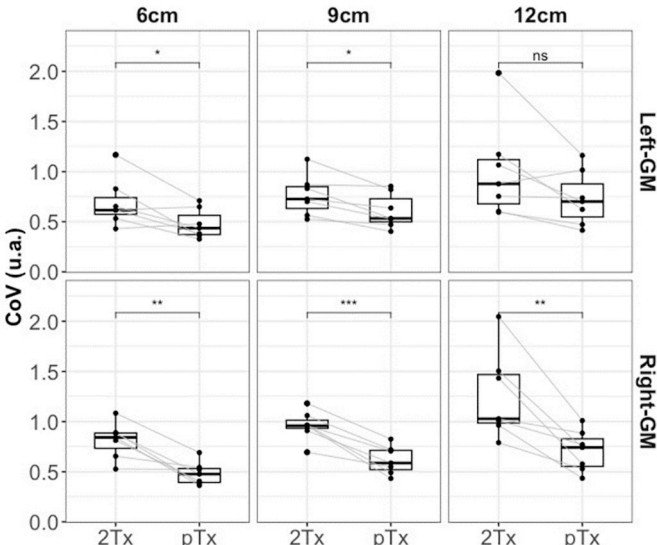

**Fig 5. Boxplot of the coefficient of variance (CoV) for both coils and all three additional slab inversion thicknesses.** The boxplot shows that for all conditions, 2Tx yielded higher CoV than pTx. Higher CoV reflects the amount of noise in the perfusion data. The lines between each coil denote each participant's CoV. A Paired t-test was used to check for statistical differences. A significant difference was observed, and the level of significance is denoted by the asterisk symbol (* represents $p < 0.01$, ** represents $p < 0.001$, *** represents $p < 0.0001$ and ns is the representation of not significant p-value).

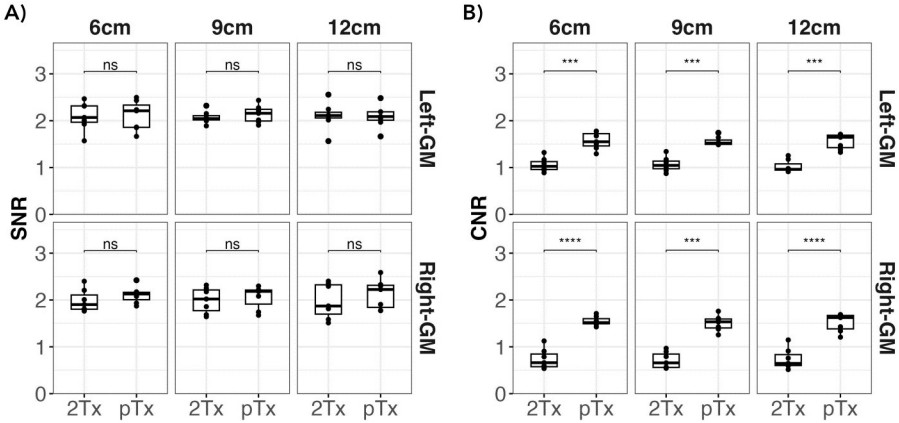

**Fig 6. Assessing signal-to-noise ratio (SNR) (Panel A) and contrast-to-noise ratio (CNR) (Panel B) across different coil configurations.** The boxplot illustrates the distribution of SNR and CNR values for the 2Tx and pTx coils. The pTx coil shows higher median SNR values (indicated by dark strokes) for both the left and right gray matter (GM), and these results are also consistent across slab inversion thicknesses. In terms of CNR, pTx yielded higher values than 2Tx. Statistical analysis using paired t-test (Bonferroni corrected) revealed a significant difference in CNR assessment for all thicknesses (p<0.00001) for both the right and left hemispheres, while no statistical difference was found in SNR. The level of significance is denoted by an asterisk symbol, with **** representing p<0.0001.

to be more spread in the 2Tx than in the pTx coil, specifically for the right GM region. Statistical assessment showed no significant differences between both coils. The CNR boxplot (Fig 6 panel B), displays the distribution of CNR across participants for both coils. We observed higher CNR for pTx than the 2Tx coil, and the paired t-test showed a significant difference (Bonferroni corrected) between both coil configurations (p<0.00001).

Regarding the fMRI experiment, robust perfusion responses in the motor cortex were obtained in all participants with both coils. Fig 7 shows the tSNR values extracted from the gray-matter mask and an exemplar tSNR map from one of the participants (P07). The individual data points show a noticeably higher tSNR for the pTx coil compared to 2Tx for all participants, albeit, not statistically significant. Fig 8 presents an illustrative slice of the mean perfusion activation map from all participants engaged in the fMRI experiment. We found comparable activation maps between 2Tx and pTx in all participants. The mean and the maximum z-values for both coils are depicted in Fig 9. There were no significant differences between the z-values obtained with the two different setups.

## Discussion

In the present study, we implemented a Pulsed FAIR ASL sequence at 7T on two different systems: a standard 2-channel transmit system with a 32Rx2Tx coil and the pTx system of the same 7T scanner (32Rx8Tx). Both systems utilized the same pulse-sequence and RF pulses. We aimed to compare the effect of the performance of both rf-coils configurations on the ASL quality.

The FAIR sequence with TR-FOCI inversion pulse yielded high image quality on both 2Tx and pTx systems. Upon increasing the thickness of the slab-selective inversion slab (thereby reducing the label bolus width), signal decreases were smaller using the pTx system, reflecting the better coil performance of the latter. We observed this behavior in Figs 2 and 3, but it is also reflected in the slightly higher slopes for the CoV vs. slab thickness in the 2Tx system compared to the pTx system in Figs 4 and 5. With the increasing thickness of the inversion slab, the effective gap between the imaging volume and the top of the labeling slab increases.

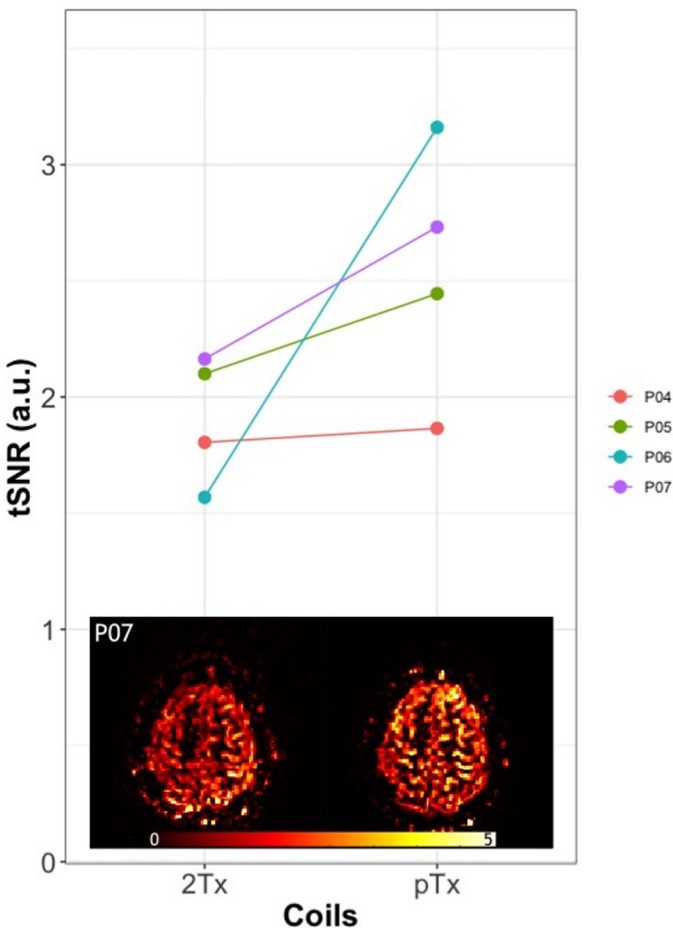

**Fig 7. Temporal Signal-to-noise ratio (tSNR) of the gray matter in the fMRI data.** On the top part of the figure, we show the scatter plots of tSNR values from both coils extracted from the gray-matter mask in the fMRI data. The lines between each coil denote each participant's tSNR signal. On the bottom part of the figure, we show a single slice of the tSNR maps from one of the participants (P07). The pTx yielded higher tSNR for all participants than the standard 2Tx coil.

Labeling will, therefore, be more dependent on more distal locations in which the $B1^+$ of the transmit coil is known to decrease. It can therefore be concluded that by using pTx the inversion becomes more efficient, especially in the lower regions.

Moreover, Figs 2 and 3 depict a significant asymmetry in perfusion signal intensity between the right and left hemispheres while using the 2Tx system. It is important to mention that this asymmetric behavior can be misleading when ASL is used for clinical purposes, as it can be perceived as a spatial perfusion variation, hypoperfusion, caused by a type of vascular impairment, like a stenosed internal carotid artery [35]. In our case, the observed difference is not driven by the overall $B_1^+$ amplitude that would result in a global decrease in signal intensity, but rather by the different patterns of $B_1^+$ for the two coils at the level of the labeling. By looking at slices of the $B1^+$ map that correspond to the imaging slices, it was confirmed that the asymmetry was not caused by local variation in the excitation flip angle: S1 Fig in S1 File shows the difference pattern of $B1^+$ maps for both coils. Interestingly, similar asymmetric behavior was previously reported for a different 7T system using a single-channel transmit coil [36]. There, the authors proposed extending their coil's coverage by adding dielectric pads [15]

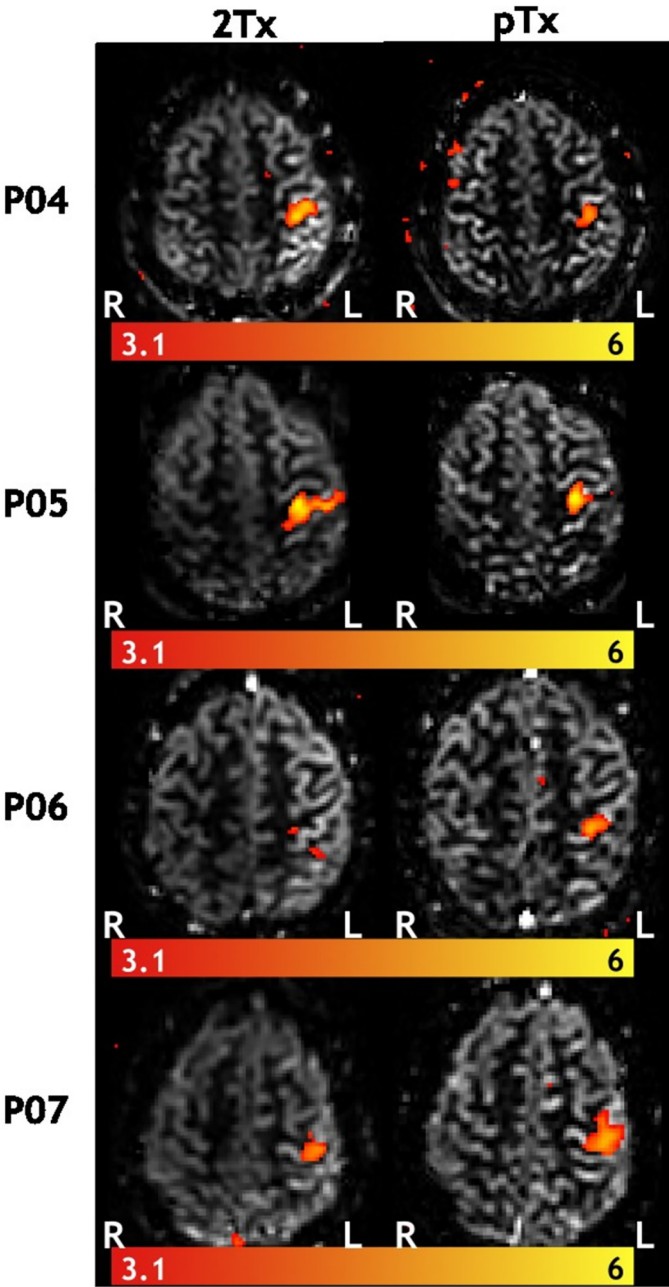

**Fig 8. A single slice of activation maps, generated from perfusion-weighted data during a right-hand finger-tapping experiment, showcases mean perfusion-weighted images superimposed with Z-Statistical maps in subject space across different slices.** Consistent activation patterns were noticeable across all participants for both coils.

to ensure a more homogenous distribution of the B1+ and, therefore, the resulting perfusion signal. The pTx coil had a similar effect, improving the B1+ distribution across the brain, including the lower brain areas where the tag is formed in FAIR.

Additionally, we also evaluated the SNR across both ROIs and found a higher median for pTx compared to the 2Tx coil; however, this discrepancy lacked statistical significance. Interestingly, the SNR distribution across participants appears more widely dispersed with the 2Tx

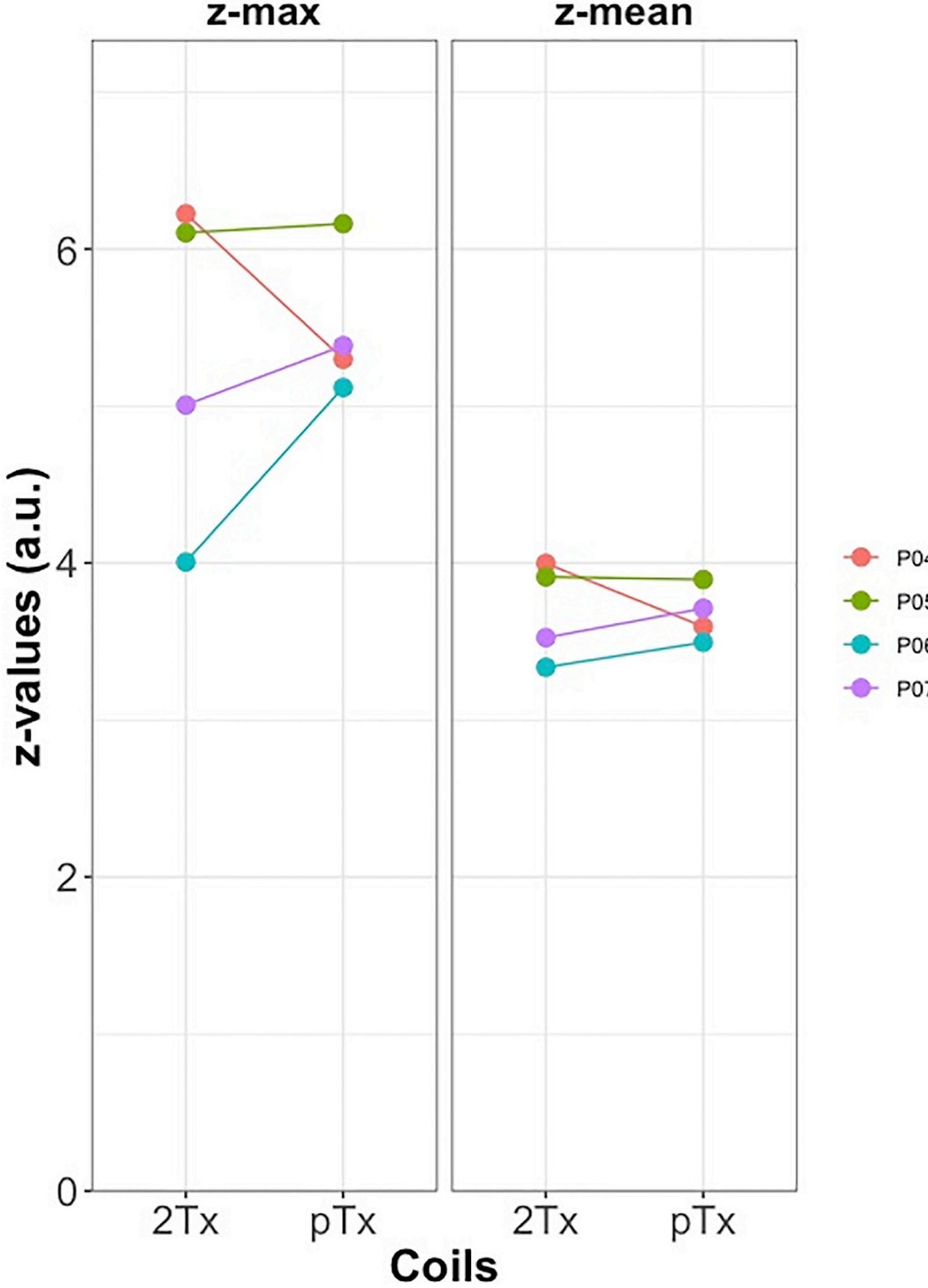

**Fig 9. Max and mean z-values from all participants of the fMRI experiment.** Z-values were extracted from Z-statistical activation maps with a z > 3.1 and p = 0.05 threshold. Participants P06 and P07 demonstrated higher mean and max z-values for pTx compared to 2Tx, while P05 exhibited a slight increase in pTx compared to 2Tx. However, P04 displayed decreased z-values (mean and max) for pTx relative to 2Tx. While our processing pipeline did not identify issues of excessive motion for participant P04, we acknowledged the possibility that the differences in the participant's engagement with the task could have influenced the observed decreased z-values while using pTx.

than the pTx coil, specifically for the right gray matter region. Given that our SNR measurements are likely driven by the signal intensities rather than the noise levels, we speculate that the observed variations have the same origin as the signal intensities that are observed between both hemispheres (see Figs 2 & 3). On the other hand, our dataset showed statistically significant differences in CNR between pTx and standard 2Tx coil. Higher CNR values indicate better differentiation between GM and WM regions, which can be crucial for lesion detection [37].

The initial evaluation of the fMRI dataset focused on tSNR. Our observations indicated an enhancement in tSNR with the use of a pTx coil as opposed to 2Tx, as demonstrated in Fig 7. These findings strongly indicate that using the pTx in CP mode (or close to CP mode, i.e., no RF-shimming or tailored RF pulses) can potentially yield superior gains in tSNR for ASL perfusion images. Given that both sequences were executed with nearly identical settings, we believe that the increase in tSNR highlights the robustness of signal intensity distribution when utilizing pTx coil. This consistency in results was also evident in experiment 1, where pTx demonstrated superior signal stability across all regions (lower CoV, CoV = 1/tSNR), including both the right and left gray matter ROIs and additional inversion slab thickness.

Previous studies have already shown an increase in tSNR in BOLD after the implementation of tailored RF pulses, like Universal pulses (UPs) or kt-points [38, 39] or just using RF $B_1^+$ shimming [40]. To date, the use of tailored pTx pulses such as kt-points and UPs in ASL literature is scarcer, especially for similar purposes as in BOLD fMRI studies, specifically targeting whole brain signal homogeneity. In principle, employing these methods in ASL fMRI could offer advantages such as increased tSNR, improved signal homogeneity, and potentially alleviating power deposition constraints. Initial insights demonstrated the benefits of these pulses in velocity-selective ASL [41]. The spatially selective ASL studies that used pTx capabilities were especially focused on enhancing labeling efficiency by improving $B_1^+$ shimming at the inflowing arteries [22–24]. This was especially important in those studies as they employed pCASL labeling for which labeling efficiency is highly dependent on B1 with SAR-restriction easily leading to prolonged TR [22–24].

Our fMRI experiment consistently elicited robust perfusion responses within the motor cortex across all participants using both coils. The perfusion activation maps demonstrate comparable activation maps between 2Tx and pTx. The mean and maximum z-scores calculation revealed higher mean and maximum z-values for pTx than 2Tx, for three out of four participants. Overall, the results of the fMRI experiment suggest that both coils can be used effectively for perfusion-based fMRI studies in the motor cortex.

## Conclusion

Acquisition of FAIR perfusion-weighted images with TR-FOCI pulse was feasible at 7T using an 8-channel pTx coil. Here, we show that utilizing a pTx coil with multiple transmit channels provides notable advantages, even when the system is used in near-quadrature mode. The pTx setup improved signal stability and homogeneity, which are crucial for enhancing the sensitivity and accuracy of perfusion-weighted measurements.

## Supporting information

**S1 File. Supplementary material.** This supplementary file presents detailed results from the linear regression analysis, systematically summarized in S1-S4 Tables. It also features S1, S2 Figs, which illustrate $B_1^+$ maps and individual perfusion maps, providing more details for both coil configurations.
(PDF)

## Author Contributions

**Conceptualization:** Ícaro Agenor Ferreira Oliveira, Matthias J. P. van Osch, Wietske van der Zwaag, Lydiane Hirschler.

**Data curation:** Ícaro Agenor Ferreira Oliveira, Robin Schnabel, Wietske van der Zwaag.

**Formal analysis:** Ícaro Agenor Ferreira Oliveira, Robin Schnabel.

**Funding acquisition:** Wietske van der Zwaag.

**Investigation:** Ícaro Agenor Ferreira Oliveira, Robin Schnabel, Matthias J. P. van Osch, Wietske van der Zwaag, Lydiane Hirschler.

**Methodology:** Ícaro Agenor Ferreira Oliveira, Robin Schnabel, Matthias J. P. van Osch, Wietske van der Zwaag, Lydiane Hirschler.

**Project administration:** Ícaro Agenor Ferreira Oliveira, Matthias J. P. van Osch, Wietske van der Zwaag, Lydiane Hirschler.

**Resources:** Wietske van der Zwaag, Lydiane Hirschler.

**Software:** Ícaro Agenor Ferreira Oliveira.

**Supervision:** Matthias J. P. van Osch, Wietske van der Zwaag, Lydiane Hirschler.

**Validation:** Ícaro Agenor Ferreira Oliveira, Robin Schnabel, Matthias J. P. van Osch, Wietske van der Zwaag, Lydiane Hirschler.

**Visualization:** Ícaro Agenor Ferreira Oliveira, Robin Schnabel, Matthias J. P. van Osch, Wietske van der Zwaag, Lydiane Hirschler.

**Writing – original draft:** Ícaro Agenor Ferreira Oliveira.

**Writing – review & editing:** Ícaro Agenor Ferreira Oliveira, Matthias J. P. van Osch, Wietske van der Zwaag, Lydiane Hirschler.

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
