## [Decision Letter · Decision Letter 0]

18 Mar 2024

PONE-D-23-37867Advancing 7T perfusion imaging by Arterial Spin Labeling: Using parallel transmit for enhanced labeling efficiency and image quality.PLOS ONE

Dear Dr. Oliveira,

Thank you for submitting your manuscript to PLOS ONE. After careful consideration, we feel that it has merit but does not fully meet PLOS ONE’s publication criteria as it currently stands. Therefore, we invite you to submit a revised version of the manuscript that addresses the points raised during the review process.

We look forward to receiving your revised manuscript.

Kind regards,

Federico Giove, PhD

Academic Editor

PLOS ONE

Journal Requirements:

2. Please note that PLOS ONE has specific guidelines on code sharing for submissions in which author-generated code underpins the findings in the manuscript. In these cases, all author-generated code must be made available without restrictions upon publication of the work. 

Please review our guidelines at https://journals.plos.org/plosone/s/materials-and-software-sharing#loc-sharing-code and ensure that your code is shared in a way that follows best practice and facilitates reproducibility and reuse.

"This work was supported by a Royal Netherlands Academy for Arts and Sciences (KNAW) research grant NWO (Nederlandse Organisatie voor Wetenschappelijk Onderzoek), Number: TTW VI.Vidi.198.016 to W.Z."

5. We note that you have indicated that there are restrictions to data sharing for this study. For studies involving human research participant data or other sensitive data, we encourage authors to share de-identified or anonymized data. However, when data cannot be publicly shared for ethical reasons, we allow authors to make their data sets available upon request. For information on unacceptable data access restrictions, please see http://journals.plos.org/plosone/s/data-availability#loc-unacceptable-data-access-restrictions. 

**Additional Editor Comments:**

Reviewers generally considered the manuscript of moderate interest because of limitations of the approach and  mild innovation.

The manuscript can be considered for publication only if methodological concerns rasied by all reviewers are fully addressed. Moreover, methodology must be described with more details and greater clarity. Appropriate quantitative measuremens should be introduced, following reviewers suggestions.

Regarding data availability, I don't share the authors view. GDPR allows publication of raw data if they are de-identified, that is if any connection between data and individual are broken. If MR images can't be associated to a person, then they are not perosnal data and are not regulated by GDPR. Of course, any personal feature (including faces in anatomical data) must be deleted.

I understand the concerns (I am an UE citizen and work in EU), but the open data principles must be adhered to. I suggest to check with the local privacy officer.

Reviewers' comments:

Reviewer's Responses to Questions

**Comments to the Author**

1. Is the manuscript technically sound, and do the data support the conclusions?

Reviewer #1: Yes

Reviewer #2: Yes

Reviewer #3: No

2. Has the statistical analysis been performed appropriately and rigorously? 

Reviewer #1: Yes

Reviewer #2: I Don't Know

Reviewer #3: No

3. Have the authors made all data underlying the findings in their manuscript fully available?

Reviewer #1: Yes

Reviewer #2: No

Reviewer #3: No

4. Is the manuscript presented in an intelligible fashion and written in standard English?

Reviewer #1: Yes

Reviewer #2: Yes

Reviewer #3: No

5. Review Comments to the Author

Reviewer #1: In this work, the authors compared the effects of different transmit coils (two-channel v.s. 8-channel) on multi-slice cerebral pulsed arterial spin labeling (pASL) at 7T. The comparison was applied to both baseline with three different inversion slab thicknesses and functional ASL of the hand region in the motor cortex with a finger-tapping functional task. The author found lower coefficient of variance (CoV) between the perfusion signal and the inversion slab thickness using 8-channel transmit than two-channel, which represents a higher robustness. The author also demonstrated higher temporal SNR and comparable capabilities for functional ASL using the 8-channel transmit coil.

The experiments in this work were well-designed. The novelty of this manuscript is moderate since pASL has not been a preferred method for brain perfusion mapping in recent years. This manuscript is well written with a clear structure and proper language. Here are some minor tips:

1. The title ‘Advancing 7T perfusion imaging by Arterial Spin Labeling: Using parallel transmit for enhanced labeling efficiency and image quality’. It is recommended to specify ‘Pulsed’ Arterial Spin Labeling. Also, ‘enhanced labeling efficiency and image quality’ is too general since there are no results involving labeling efficiency as a metric. It is better to specify ‘higher robustness’ and ‘higher temporal SNR’. Besides, the running title does not even mention ‘parallel transmit’, which is the key of this manuscript.

2. ‘Perfusion’ labeled in the color bar of Figure 2 is not appropriate, which should be a normalized value such as ml/100g/min. It is better labeled as ‘Perfusion signal’ or ‘Perfusion weighted signal’. Besides, Figure 3 should also come with a color bar.

Reviewer #2: This study investigates the effects of two transmit coils configuration on ASL perfusion imaging with FAIR using TR-FOCI for labeling and background suppression. The two configurations were: a) a parallel transmit system (pTX) with 32 Rx and 8 Tx channels, and b) a standard setup of 32Rx and 3Tx channels. This work showed that a pTx configuration is advantageous in ASL imaging at ultra-high magnetic fields given its less inhomogenous B1 field; however, CBF quantification is missing which would have given a more illustrative example of why it is better to use a pTx system. The two configurations were also tested in functional imaging. Both coils yielded robust fMRI measurements, and the tSNR was higher using the pTX system.

Major Issues

1. To validate that a pTX setting was better than the standard system two quantitative measures were used: coefficient of variance (CoV) and temporal signal-to-noise ratio (SNR); however, as shown in equation two these two measurements are dependent from each other (or basically the same measurement i.e., CoV = 1/tSNR). CBF maps should be quantified in order to add robustness to the study.

2. Line 74: it would also allow for higher perfusion imaging resolution as seen in https://doi.org/10.1371/journal.pone.0215998

3. Line 152: I would show here in a figure the coils b1 maps or give a reference to the b1 maps showed in the supplementary material. Also, for this figure in the supplementary materials the scale and color bar are missing.

4. Was whole brain coverage achieved? If yes, it will be worth to show those images, or a comparison of maximum brain coverage achieved with both systems (pTx versus standard), as whole brain coverage is also specified in the introduction as one of the advantages of acquiring ASL at ultra high magnetic fields.

5. What correction method for multiple comparisons was used for the fMRI study?

6. Line 233 : were t-test comparisons corrected for multiple comparisons?

7. Line 240 in the caption of figure 4 caption: the wording of two different ROIs is confusing; these are the same right/left hemisphere gray matter masks, right?

8. Line 255 : was the tSNR significant higher for the pTX coil compared to the 2Tx coil?

9. Line 272: it might be worth to address in the discussion why do you think P04 displayed decreased z-values for pTX relative to 2Tx (something that was not really expected).

10. Caption in Figure 1 of the supplementary material does not specify the preinversion thickness used for these FAIR acquisitions.

Minor issues

1. Line 22: A in arterial spin labeling should not be capitalized. Overall, there is no consistency between capitalized and non-capitalized description of abbreviations, e.g., (line 27) flow-sensitive alternating inversion recovery (FAIR) or (line 28) Time-Resolved Frequency Offset corrected Inversion (TR-FOCI). Please choose a style and keep it constant throughout the manuscript.

2. Line 45: BOLD fMRI has not been previously described or spelled out.

3. References are missing in line 63 and 65.

4. Line 66: It is recommended not to use contractions in formal writing (i.e., ASL’s)

5. Line 145: C in Control should be in low case

6. Line 170: G in Gray-matter should be in low case. These style errors are constant through-out the document please correct.

7. Line 231: I think the sentence should read: …, meaning that there is more perfusion signal loss at larger inversion slab widths in the 2Tx compared to the pTx coil.

8. Line 298: add citation

9. Line 302-308: very interesting

10. Line 317: capitalize ROIs

11. Line 319-320: UP definition is missing.

Reviewer #3: This manuscript presents a pTx implementation of a pulsed arterial spin labeling technique at ultra-high field for brain perfusion imaging. They demonstrate a comparison of two coil system setups: 2 channel transmit and 8 channel transmit pTx. Two different experiments were performed to evaluate the coil performances. Authors claim the pTx system produces a more homogenous transmit field which translated into improved perfusion signal acquisition.

I agree that ASL can greatly benefit from ultra-high field strength. The use of pTx is becoming a popular approach to combat the challenges which are more evident at 7T. However, there are several technical problems and gaps in the proposed methodology which lack clarity and are of major concern. This leads to many assumptions which readers may need to make. It appears the manuscript focuses more on a coil comparison rather than the potential capabilities of pTx as implied in the title. It is unclear where the capabilities of using pTx are showcased beyond using the 8Tx pTx coil. I suggest the following concerns are addressed prior to recommendation of publication.

Major comments:

1. There are various ASL labeling schemes which can be broadly categorized as spatially selective or velocity selective. Select statements made in the introduction regarding spatial coverage and dependence on arterial transit times is generally only applicable to spatially selective ASL schemes (PASL and pCASL). It will be insightful to elaborate and acknowledge in the introduction other ASL techniques which exist that are not spatially selective such as velocity selective ASL and their progress with using pTx.

2. Line 118 – In section 2.2 Data acquisition, it is briefly mentioned that B1-shimming was implemented over the entire brain. I understand separate volunteer data was used to achieve the B1-shim however there is ambiguity in the description. Please provide more detail on the size of volunteer data to calculate the shim parameters and whether this was a universal B1-shim applied for all subjects scanned for the study. Did you consider using a subject tailored shim or tailored pTx pulses for better B1 correction? If not, why? Please also specify in what aspect of the full ASL FAIR sequence was this shim applied to?

3. Line 120 – This next point follows on from comment 2. It is unclear where pTx capabilities is utilized despite the use of a pTx coil. The authors describe adiabatic tr-FOCI pulses are used for the labeling and two background suppression pulses. Adiabatic pulses are insensitive to B1 variation. They don’t require a pTx coil to be implemented effectively. My understanding is that these adiabatic pulse elements is what contributes significantly to the efficiency of the labeling and as a result the perfusion signal. This then only leaves the 2D-EPI excitation pulses and saturation modules. It is not mentioned and therefore unclear where exactly the pTx was applied to benefit the ASL technique evaluated.

4. Line 152 – Was there any verification in the setup to ensure a fair comparison of the two coils? Authors mentioned a different maximum B1 limit was applied for the two coils respectively. There is a significant difference between the maximum B1 limit. This leads me to assume significantly lower/capped reference voltage was used for the 2Tx coil. Using a lower reference voltage may result in an overall lower flip angle distribution. This appeared to translate in the B1 maps shown in the supplementary material. Could you explain how experiments were performed to give a fair comparison. It would be ideal to consider a baseline calibration of the reference voltage prior to acquisition of perfusion protocols.

5. Line 291 – The authors state that labeling is dependent on the distal location selected. I agree that this has an effect as there are associated blood transition times and T1 relaxation effects. However, they conclude pTx is more efficient for inversion. My understanding is adiabatic pulses were used for inversion. When set up to satisfy adiabatic conditions they are insensitive to B1 variation. This leads me to question whether the tr-FOCI pulses were optimised and whether the restriction of maximum B1 limit influenced the overall inversion fidelity of the pulses?

6. It is not easily understandable what the cause of the asymmetry that occurs in your acquired 2Tx data. You refer to the B1 maps supplied in the supplementary material and suggest that the cause is not due to excitation distribution instead it is the RF profile. My understanding of “RF profile of the coils” is that it relates to transmit sensitivity which is the generated B1 distribution. It is unclear whether the right-side bias is caused by coil geometric sensitivities, transmit efficiency or the B1-shim applied. Can you explain what you mean by “RF profile of the coils” incase I have misunderstood.

7. Compared to other regions of the brain, the B1 in the motor cortex at 7T is relatively adequate. This may be one contributing explanation why the fMRI results were comparable between the two coil setups. Did you consider investigating and comparing other areas (e.g. visual cortex or cerebellum) where B1 is less sufficient to showcase the benefits/diversity of pTx?

Minor comments:

8. Line 77 – There may be more suitable references to include that demonstrates comparative implementation of different adiabatic pulses for PASL.

- Zimmer F, O'Brien K, Bollmann S, Pfeuffer J, Heberlein K, Barth M. Pulsed arterial spin labelling at ultra-high field with a B 1 (+) -optimised adiabatic labelling pulse. MAGMA. 2016 Jun;29(3):463-73. doi: 10.1007/s10334-016-0555-2. Epub 2016 Apr 15. PMID: 27084187.

- Wang K, Shao X, Yan L, Ma SJ, Jin J, Wang DJJ. Optimization of adiabatic pulses for pulsed arterial spin labeling at 7 tesla: Comparison with pseudo-continuous arterial spin labeling. Magn Reson Med. 2021 Jun;85(6):3227-3240. doi: 10.1002/mrm.28661.

9. Line 88 – Wording is confusing as two different concepts are trying to be conveyed here. SAR constraints are stringent at UHF whereas sequences can be less SAR intensive compared to others. PASL uses multiple adiabatic pulses which are SAR intensive. Could you please elaborate why you claim PASL may be less SAR intensive compared to other ASL techniques?

10. Figure 1 shows the coverage and location of the labeling and imaging regions. It would benefit to show a full sequence diagram with all sequence elements.

11. It may be insightful to detail the differences in coil configuration and geometry where appropriate since limited pTx pulse capabilities are implemented.

12. Line 299 – Could you please explain what you mean by “calibration of B1”.

13. Line 302 – Wording implies there is a difference map calculated between the two independently acquired B1 maps of the two coils.

14. Line 320 – There was initial insights demonstrated in the benefits of using tailored pTx pulses for velocity selective labelling at 7T which are directly applicable to ASL techniques.

- Wu C-Y, Jin J, Dixon C, Maillet D, Barth M, Cloos MA. Velocity selective spin labeling using parallel transmission. Magn Reson Med. 2024; 91(4): 1576-1585. doi: 10.1002/mrm.29955

6. PLOS authors have the option to publish the peer review history of their article (what does this mean?). If published, this will include your full peer review and any attached files.

Reviewer #1: No

Reviewer #2: **Yes: **Maria G. Mora Alvarez

Reviewer #3: No

---

## [Author Response · Author response to Decision Letter 0]

8 May 2024

Journal requirements

1. Response: We have now formatted the manuscript accordingly. 

2. All in-house scripts developed for this study can be accessed via the provided GitHub page (https://github.com/icaroafoliveira/ASL_7T_pTx_scripts).

3. Thank you for pointing this out. We fixed that in the rebuttal submission. 

4. Additional information about the financial disclosure was added to the cover letter. "The funders had no role in study design, data collection and analysis, decision to publish, or preparation of the manuscript."

5. Unfortunately, this is not our view, but a legal restriction explained to us first by the privacy officer of the Royal Academy of Arts and Sciences (KNAW). To clarify this further, we have contacted a legal advisor who has drawn up an advice for the KNAW on (f)MRI scans and the AVG (the Dutch name for GDPR). They state clearly that brain scans are unique, like fingerprints and should be considered personal information as this unique information can be used for identification, even after defacing. All future data sharing should be covered by a data sharing agreement between recipient and the KNAW. Requests can be sent to w.vanderzwaag@spinozacentre.nl. 

 

Response to reviewers

Reviewer #1: In this work, the authors compared the effects of different transmit coils (two-channel v.s. 8-channel) on multi-slice cerebral pulsed arterial spin labeling (pASL) at 7T. The comparison was applied to both baseline with three different inversion slab thicknesses and functional ASL of the hand region in the motor cortex with a finger-tapping functional task. The author found lower coefficient of variance (CoV) between the perfusion signal and the inversion slab thickness using 8-channel transmit than two-channel, which represents a higher robustness. The author also demonstrated higher temporal SNR and comparable capabilities for functional ASL using the 8-channel transmit coil.

The experiments in this work were well-designed. The novelty of this manuscript is moderate since pASL has not been a preferred method for brain perfusion mapping in recent years. This manuscript is well written with a clear structure and proper language. Here are some minor tips:

We thank the reviewer for the time taken to review the present manuscript. We appreciate all positive comments and suggestions. In the context of moderate field strengths (1.5 and 3T), we agree with the reviewer that pulsed ASL has not been a preferred method for brain perfusion mapping. However, as the use of the preferred pCASL method is challenging at 7T, pulsed ASL remains a popular method at high-field MRI [1] as well for functional ASL experiments in which the shorter temporal footprint of PASL is an advantage over pCASL. We therefore believe that this method is still relevant for the ultra-high field community, especially in the context of functional MRI (fMRI). Please find the detailed answers to the reviewer’s comments below.

1. The title ‘Advancing 7T perfusion imaging by Arterial Spin Labeling: Using parallel transmit for enhanced labeling efficiency and image quality’. It is recommended to specify ‘Pulsed’ Arterial Spin Labeling. Also, ‘enhanced labeling efficiency and image quality’ is too general since there are no results involving labeling efficiency as a metric. It is better to specify ‘higher robustness’ and ‘higher temporal SNR’. Besides, the running title does not even mention ‘parallel transmit’, which is the key of this manuscript.

Response: We agree with the reviewer about the generalisability of our current manuscript title. We changed the title to “Advancing 7T Perfusion Imaging by Pulsed arterial spin labeling: Using a parallel transmit coil for enhanced robustness and temporal SNR”.

2. ‘Perfusion’ labeled in the color bar of Figure 2 is not appropriate, which should be a normalized value such as ml/100g/min. It is better labeled as ‘Perfusion signal’ or ‘Perfusion weighted signal’. Besides, Figure 3 should also come with a color bar.

Response: We changed the label to �M, with �M defined as the subtraction of control and label images (∆M=S_control-S_label). We added this notion to the manuscript. For Fig 3, a color bar was added.

Changes:

FAIR perfusion-weighted maps were generated through pairwise subtraction of control and label images (∆M=S_control-S_label). 

Figure changes:

Fig 2: Arterial spin labeling difference image (�M) from a single participant (P06), highlighting differences between the 2Tx and pTx systems. The pTx system generally shows more homogeneous signals (e.g. fewer right-left differences) than the 2Tx. The increased thickness of the slab-selective inversion (6cm, 9cm, and 12cm) resulted in an intensity reduction for both the 2Tx and pTx systems.

Fig 3: Single slice (#6) arterial spin labeling difference (�M) images from all 7 participants. An asymmetric right-left signal pattern can be seen on the 2Tx images for all slab thicknesses and participants. A more homogenous pattern is consistently observed in the pTx images. The contrast window is the same for all perfusion-weighted images.

 

Reviewer #2: This study investigates the effects of two transmit coils configuration on ASL perfusion imaging with FAIR using TR-FOCI for labeling and background suppression. The two configurations were: a) a parallel transmit system (pTX) with 32 Rx and 8 Tx channels, and b) a standard setup of 32Rx and 3Tx channels. This work showed that a pTx configuration is advantageous in ASL imaging at ultra-high magnetic fields given its less inhomogenous B1 field; however, CBF quantification is missing which would have given a more illustrative example of why it is better to use a pTx system. The two configurations were also tested in functional imaging. Both coils yielded robust fMRI measurements, and the tSNR was higher using the pTX system.

We thank the reviewer for all the positive feedback and valuable suggestions. In the second part of the first comment below we addressed the reviewer’s concern regarding the CBF quantification, and we have also addressed all other concerns raised by the reviewer.

Major Issues

1. To validate that a pTX setting was better than the standard system two quantitative measures were used: coefficient of variance (CoV) and temporal signal-to-noise ratio (SNR); however, as shown in equation two these two measurements are dependent from each other (or basically the same measurement i.e., CoV = 1/tSNR). CBF maps should be quantified in order to add robustness to the study.

Response R2.C1.1 (metrics):

The main assessment of the study was the temporal stability through CoV and tSNR, CoV was used to assess the temporal stability over the different slab inversion thicknesses, and the tSNR to assess the fMRI dataset. To further improve the robustness of the study, we now incorporated two additional metrics: signal-to-noise ratio (SNR) and contrast-to-noise ratio (CNR). 

Changes:

Data analysis section 

Additionally, signal-to-noise ratio (SNR) and contrast-to-noise ratio (CNR) were also assessed. The SNR was calculated using equation [3], and it was defined as the mean divided by the standard deviation of the gray-matter mask. The CNR was calculated using equation [4], �WM and �GM are the mean signal intensities of white matter and gray matter regions of interest, respectively. �GM and �WM are the standard deviation of the signal intensities in the gray matter and white matter regions of interest, respectively.

SNR= μ_GM⁄σ_GM [3]

CNR= abs(μ_GM-μ_WM )/√((σ_GM^2+σ_WM^2 ) ) [4]

Results section 

To assess the level of noise between the two coils, we investigated the SNR and CNR. For the SNR the left and the right gray matter ROIs were assessed. As can be seen in Fig 6 panel A, the median SNR was higher for pTx compared to the standard 2Tx coil in both ROIs and across the different slab inversion thicknesses. The SNR distribution across participants seems to be more spread in the 2Tx than in the pTx coil, specifically for the right GM region. Statistical assessment showed no significant differences between both coils. The CNR boxplot (Fig 6 panel B), displays the distribution of CNR across participants for both coils. We observed higher CNR for pTx than the 2Tx coil, and the paired t-test (Bonferroni corrected) showed a significant difference between both coil configurations (p<0.00001). 

Discussion section

Additionally, we also evaluated the SNR across both ROIs and found a higher median for pTx compared to the 2Tx coil, however, this discrepancy lacked statistical significance. Interestingly, the SNR distribution across participants appears more widely dispersed with the 2Tx than the pTx coil, specifically for the right gray matter region. Given that our SNR measurements are likely driven by the signal intensities rather than the noise levels, we speculate that the observed variations have the same origin as the signal intensities that are observed between both hemispheres (see Figs 2&3). On the other hand, our dataset showed statistically significant differences in CNR between pTx and standard 2Tx coil. Higher CNR values indicate better differentiation between GM and WM regions, which can be crucial for lesion detection [2].

Fig 6: Assessing signal-to-noise ratio (SNR) (Panel A) and contrast-to-noise ratio (CNR) (Panel B) across different coil configurations. The boxplot illustrates the distribution of SNR and CNR values for the 2Tx and pTx coils. The pTx coil shows higher median SNR values (indicated by dark strokes) for both the left and right gray matter (GM), and these results are also consistent across slab inversion thicknesses. In terms of CNR, pTx yielded higher values than 2Tx. Statistical analysis using paired t-test (Bonferroni corrected) revealed a significant difference in CNR assessment for all thicknesses (p<0.00001), while no statistical difference was found in SNR. The level of significance is denoted by an asterisk symbol, with **** representing p<0.0001.

Response R2C1.2 (CBF quantification): 

While we appreciate the reviewer’s suggestion regarding the quantification of CBF maps to enhance the robustness of our study, we respectfully disagree with this statement. Our study specifically focuses on comparing the performance of two distinct coils at 7T, by evaluating the differences between these coils in terms of image quality (tSNR, SNR, and CNR). By not quantifying CBF, we aim to highlight the potential differences in coil variations, which could be masked or normalized through quantification (i.e. make the difference insensitive to coil variations). Thus, while we acknowledge the importance of robust quantification in certain contexts, we believe that in the scope of our study, the omission of CBF quantification allows for a more direct comparison of coil performance, thereby providing valuable insights into the practical implications of using different coils at 7T.

Changes:

FAIR perfusion-weighted maps were generated through pairwise subtraction of control and label images (∆M=S_control-S_label). It is worth mentioning that while we acknowledge the importance of robust quantification in certain contexts, in the scope of this study, the omission of CBF quantification allows for a more direct comparison of coil performance. Hereby, we adhere to the principle of staying as close to the measured data as possible when analyzing the quality of the data.

2. Line 74: it would also allow for higher perfusion imaging resolution as seen in https://doi.org/10.1371/journal.pone.0215998

Response R2.C2:

We added the suggested reference.

Changes:

Moreover, it would also allow longer readout acquisitions, increased brain coverage, as well as higher spatial resolution perfusion images [3].

3. Line 152: I would show here in a figure the coils b1 maps or give a reference to the b1 maps showed in the supplementary material. Also, for this figure in the supplementary materials the scale and color bar are missing.

Response R2.C3:

The DREAM B1 map is the scanner-generated map. The maps are normalized. Thank you for pointing out the missing scale and color bar, which we now added. We also added a reference to the B1 maps in the supplementary material. Now referred to as S1 Fig.

S1 Fig

4. Was whole brain coverage achieved? If yes, it will be worth to show those images, or a comparison of maximum brain coverage achieved with both systems (pTx versus standard), as whole brain coverage is also specified in the introduction as one of the advantages of acquiring ASL at ultra high magnetic fields.

Response R2.C4:

While ultra-high magnetic field strengths do facilitate larger brain coverage, it is important to note that we did not explore whole brain coverage protocols in the present study. This was mainly driven by our focus on functional ASL applications for which a more focused FOV is acceptable, specifically when it would allow higher temporal resolution.

5. What correction method for multiple comparisons was used for the fMRI study?

Response R2.C5:

We did not conduct a voxel-wise multiple comparison analysis as part of the general linear model (GLM) fit. Instead, within FSL FEAT, we utilize a cluster threshold approach, with a specified z threshold equal to 3.1, as indicated in the manuscript. All the analyses were performed at the individual level and based on ROIs. To address multiple comparisons, we used Bonferroni correction for these assessments.

6. Line 233 : were t-test comparisons corrected for multiple comparisons?

Response R2.C6:

Yes, all t-tests were corrected for multiple comparisons with Bonferroni correction. We added this notion throughout the manuscript.

Methods section:

Changes: The statistical assessment was conducted in R version 4.2.1 [4], using t-tests with alternative hypotheses that the means (pTx vs 2Tx) are different. To address multiple comparisons, we applied Bonferroni correction.

Results section:

Changes: A paired t-test (Bonferroni corrected) revealed a significant difference in both hemispheres for the two labeling thicknesses, 6cm, and 9cm.

7. Line 240 in the caption of figure 4 caption: the wording of two different ROIs is confusing; these are the same right/left hemisphere gray matter masks, right?

Response R2.C7:

Thank you for pointing this out. We changed the caption accordingly.

Changes:

Fig.4 Caption

Coefficient of variance (CoV) vs. additional (label thickness) slab inversion thickness for right and left masks and coils (2Tx and pTx)…

8. Line 255 : was the tSNR significant higher for the pTX coil compared to the 2Tx coil?

Response R2.C8:

Text in line 255 refers to the fMRI dataset, and for this dataset, we did not find a statistical significance for the alternative hypothesis of tSNR being different for parallel transmit (pTx) compared to standard 2Tx (p-value=0.15). We explicitly stated this notion in the manuscript.

Changes:

… albeit, not statistically significant

9. Line 272: it might be worth to address in the discussion why do you think P04 displayed decreased z-values for pTX relative to 2Tx (something that was not really expected).

Response R2.C9:

We thank the reviewer for this question. While our processing pipeline did not identify issues of excessive motion, we acknowledged the possibility that the differences in the participant’s engagement with the task could have influenced the observed decreased z-values while using pTx coil.

Changes:

Fig.9 Caption

While our processing pipeline did not identify issues of excessive motion for participant P04, we acknowledge the possibility that the differences in the participant’s engagement with the task could have influenced the observed decreased z-values while using pTx.

10. Caption in Figure 1 of the supplementary material does not specify the preinversion thickness used for these FAIR acquisitions.

Response R2.C10:

Thank you for pointing this out. The maps shown are from the 6 cm slab inversion thickness. 

Changes:

S2 Fig: Six slices (from 6-11) of perfusion-weighted ASL from all seven participants highlighting differences between the 2Tx and pTx systems. The pTx system generally shows a more homogeneous signal (e.g. fewer right-left differences) than the 2Tx. The

---

## [Decision Letter · Decision Letter 1]

9 Jul 2024

PONE-D-23-37867R1Advancing 7T perfusion imaging by pulsed arterial spin labeling: Using a parallel transmit coil for enhanced labeling robustness and temporal SNR.PLOS ONE

Dear Dr. Oliveira,

Thank you for submitting your manuscript to PLOS ONE. After careful consideration, we feel that it has merit but does not fully meet PLOS ONE’s publication criteria as it currently stands. Therefore, we invite you to submit a revised version of the manuscript that addresses the points raised during the review process.

The work is substantially improved and the authors did their best to take care of the concerns raised by reviewers and to address limitations. There are still some comments, that can be addressed in a minor revision. Regarding the data sharing, if the local privacy officier has this restrictive interpretation of GDPR, authors must take care, in future experiments, to explicitily seek consent by particiapnt to share raw data. In future, limited access to raw data will not be acceptable to PLOS One (and to most funding bodies)

We look forward to receiving your revised manuscript.

Kind regards,

Federico Giove, PhD

Academic Editor

PLOS ONE

Journal Requirements:

Reviewers' comments:

Reviewer's Responses to Questions

**Comments to the Author**

1. If the authors have adequately addressed your comments raised in a previous round of review and you feel that this manuscript is now acceptable for publication, you may indicate that here to bypass the “Comments to the Author” section, enter your conflict of interest statement in the “Confidential to Editor” section, and submit your "Accept" recommendation.

Reviewer #1: (No Response)

Reviewer #2: (No Response)

Reviewer #3: (No Response)

2. Is the manuscript technically sound, and do the data support the conclusions?

Reviewer #1: Yes

Reviewer #2: Yes

Reviewer #3: Yes

3. Has the statistical analysis been performed appropriately and rigorously? 

Reviewer #1: Yes

Reviewer #2: Yes

Reviewer #3: I Don't Know

4. Have the authors made all data underlying the findings in their manuscript fully available?

Reviewer #1: Yes

Reviewer #2: No

Reviewer #3: No

5. Is the manuscript presented in an intelligible fashion and written in standard English?

Reviewer #1: Yes

Reviewer #2: Yes

Reviewer #3: Yes

6. Review Comments to the Author

Reviewer #1: Thank you for revising your manuscript and including the requested information. My previous concerns were all well addressed but there are a few more comments for the added contents (The following line numbers refer to the manuscript with tracked changes).

1. Line 110: ‘A FAIR Arterial Spin Labeling sequence was implemented using TR-FOCI RF-pulses [25] with 13ms of pulse duration, nominal flip angle of 2576 degrees, and B1 = 15μT’; Line 143: ‘For the pTx system, a maximum B1+ value of 18μT was used for the excitation pulses’; Line 144: ‘For the 2Tx, a maximum B1 of 12μT was used for the excitation pulses to accommodate amplifier limitations’. Is there a specific reason why the excitation pulses used different B1+ values for the two coils while TR-FOCI RF-pulses used another but the same maximum B1+ value?

2. Line 186: ‘μGM and μWM are the mean signal intensities of white matter and gray matter regions of interest, respectively’ should be ‘of gray matter and white matter regions of interest, respectively.

3. How were the white matter masks generated?

4. Line 197 Equation 4: the CNR is typically defined as abs(S_GM-S_WM)/σ. Here the author tried to use the noise information from both the grey matter and the white matter. I expect the denominator to be sqrt((σ_GM^2+ σ_WM^2)/2) rather than be sqrt(σ_GM^2+ σ_WM^2). Nevertheless, this is only a scaling and will not affect the outcomes.

5. Figure 6B: why were the right and left hemispheres not differentiated for the CNR boxplots?

Reviewer #2: I appreciate the authors addressing my previous comments. This study investigates the effects of two transmit coils configuration on ASL perfusion imaging with FAIR using TR-FOCI for labeling and background suppression. The two configurations were: a) a parallel transmit system (pTX) with 32 Rx and 8 Tx channels, and b) a standard setup of 32Rx and 2Tx channels. This work showed that a pTx configuration is advantageous in ASL imaging at ultra-high magnetic fields given its less inhomogeneous B1 transmit field. Additional, CoV, CNR and SNR measurements were assessed to add robustness to the study. The two configurations were also tested in functional imaging. Both coils yielded robust fMRI measurements, but the tSNR was higher using the pTX system.

Major Issues

1. I think it is important to clarify why SNR was not calculated using the standard deviation of noise in the background (air). Presumably, this is why no statistically significant difference was seen in the SNR assessment between the two coils. I know the use of parallel imaging and transmit coils make this measurement complex, but it will be worth to comment on that.

Minor issues

1. Line 49: This sentence is a phrase and not a complete sentence, it should be connected to the next sentence with a conjunction, in this case using not only…, but also… e.g., Scanning at a higher field strength not only holds the key to improving the intrinsic low SNR of ASL, but also recent research has...

2. Line 341: The should be a semicolon before however instead of a comma, as it is being used as a conjunction.

3. Line 369: I think the flow of the paragraph will be improved if this sentence is start as.. Our fMRI experiment .. since you are previously describing other studies it can be a bit confusing when reading it at a first glance as The fMRI experiment…

Reviewer #3: Thank you for the corrections that you have made in this revision. The authors have mostly addressed my comments satisfactorily.

1. (R3.C5) There is still a concern on clarity with regards to the achieved B1+=15uT for the TR-FOCI RF pulses. The authors clarified that the pulse attributes were kept consistent across both coils. However, there is a discrepancy in the manuscript (line 144) which states, “For the 2Tx, a maximum B1+ of 12uT was used for the excitation pulses to accommodate amplifier limitations”. Given this amplifier limitation, how is the B1+=15uT achieved on the 2Tx?

2. (R3.C6) Is it possible to show multiple slices of the B1 maps (S Fig. 1) which covers the labeling regions highlighted in Figure 1 to verify the claims of potential differences of the B1+ profile of the coil at the level of the tag?

Finally, there are some minor inconsistencies with abbreviations (e.g. B1+ vs B1) used throughout the manuscript. Otherwise, I’m quite satisfied with the authors’ response.

7. PLOS authors have the option to publish the peer review history of their article (what does this mean?). If published, this will include your full peer review and any attached files.

Reviewer #1: No

Reviewer #2: **Yes: **María Guadalupe Mora Álvarez

Reviewer #3: No

---

## [Author Response · Author response to Decision Letter 1]

6 Aug 2024

Journal requirements

We have updated the list of references in the manuscript.

Response to reviewers

Reviewer #1: Thank you for revising your manuscript and including the requested information. My previous concerns were all well addressed but there are a few more comments for the added contents (The following line numbers refer to the manuscript with tracked changes).

1. Line 110: ‘A FAIR Arterial Spin Labeling sequence was implemented using TR-FOCI RF-pulses [25] with 13ms of pulse duration, nominal flip angle of 2576 degrees, and B1 = 15μT’; Line 143: ‘For the pTx system, a maximum B1+ value of 18μT was used for the excitation pulses’; Line 144: ‘For the 2Tx, a maximum B1 of 12μT was used for the excitation pulses to accommodate amplifier limitations’. Is there a specific reason why the excitation pulses used different B1+ values for the two coils while TR-FOCI RF-pulses used another but the same maximum B1+ value?

Response: We thank the reviewer for the positive comments and suggestions. RF power is chosen based on the ability to run the sequence with a particular setting, with power deposition (SAR), transmit coil and amplifier properties being the typical constraints. When a potential SAR or power limit is encountered, the amplitude of the excitation pulses is reduced, keeping the amplitude of contrast-generating pulses, like inversion pulses constant. In our case, the ASL-FAIR has numerous RF pulses that can significantly increase the power deposition. Additionally, the pTx coil achieves decreased SAR, allowing for higher maximum B1+ values compared to the 2Tx coil.

2. Line 186: ‘μGM and μWM are the mean signal intensities of white matter and gray matter regions of interest, respectively’ should be ‘of gray matter and white matter regions of interest, respectively.

Changed: “mean signal intensities of gray matter and white matter regions of interest, respectively.”

3. How were the white matter masks generated?

Response: We created the white matter binary masks utilizing the white matter tissue map generated through the SPM12 segmentation.

Changes: “To create the gray-matter and white-matter masks, we used the segmentation tool from SPM12 on the averaged perfusion-weighted images. Gray and white matter binary masks for the left and right hemispheres were created manually using iTK-SNAP by editing the segmented gray-matter and white-matter masks.”

4. Line 197 Equation 4: the CNR is typically defined as abs(S_GM-S_WM)/σ. Here the author tried to use the noise information from both the grey matter and the white matter. I expect the denominator to be sqrt((σ_GM^2+ σ_WM^2)/2) rather than be sqrt(σ_GM^2+ σ_WM^2). Nevertheless, this is only a scaling and will not affect the outcomes.

Response: We agree with the reviewer’s comment. The CNR equation we used was based on the extension developed by the MRIQC pipeline (Esteban et al., 2017) from fMRIprep. We have added this notion to the manuscript.

Changes: “Both SNR and CNR equations are derived from the publicly available quality assurance pipeline (MRIQC).”.

5. Figure 6B: why were the right and left hemispheres not differentiated for the CNR boxplots?

Response: We have updated Figure 6B to differentiate the right and left hemispheres. As shown in the updated figure below, the results were very similar when separated by right and left hemispheres. 

Fig 6: Assessing signal-to-noise ratio (SNR) (Panel A) and contrast-to-noise ratio (CNR) (Panel B) across different coil configurations. The boxplot illustrates the distribution of SNR and CNR values for the 2Tx and pTx coils. The pTx coil shows higher median SNR values (indicated by dark strokes) for both the left and right gray matter (GM), and these results are also consistent across slab inversion thicknesses. In terms of CNR, pTx yielded higher values than 2Tx. Statistical analysis using paired t-test (Bonferroni corrected) revealed a significant difference in CNR assessment for all thicknesses (p<0.00001) for both the right and left hemispheres, while no statistical difference was found in SNR. The level of significance is denoted by an asterisk symbol, with **** representing p<0.0001.

 

Reviewer #2: I appreciate the authors addressing my previous comments. This study investigates the effects of two transmit coils configuration on ASL perfusion imaging with FAIR using TR-FOCI for labeling and background suppression. The two configurations were: a) a parallel transmit system (pTX) with 32 Rx and 8 Tx channels, and b) a standard setup of 32Rx and 2Tx channels. This work showed that a pTx configuration is advantageous in ASL imaging at ultra-high magnetic fields given its less inhomogeneous B1 transmit field. Additional, CoV, CNR and SNR measurements were assessed to add robustness to the study. The two configurations were also tested in functional imaging. Both coils yielded robust fMRI measurements, but the tSNR was higher using the pTX system.

Major Issues:

1. I think it is important to clarify why SNR was not calculated using the standard deviation of noise in the background (air). Presumably, this is why no statistically significant difference was seen in the SNR assessment between the two coils. I know the use of parallel imaging and transmit coils make this measurement complex, but it will be worth to comment on that.

Response: We thank the reviewer for the positive comments and suggestions. Using parallel imaging can influence the statistical and spatial distribution of noise. The noise distribution in parallel imaging is described by the spatially varying geometry factor (g-factor) and depends on parameters such as the coil geometry, phase encoding direction, and acceleration factor (Dietrich et al., 2007), and the calculation of the SNR using the same ROIs is commonly suggested as an alternative approach (Dietrich et al., 2007), the approach used in the present study was based on the quality assurance pipeline (MRIQC) (Esteban et al., 2017), which is also part of the fMRIprep quality check pipeline. We added this notion to the manuscript.

Changes: The SNR equation using the same ROI is based on a previous recommendation (Dietrich et al., 2007) to minimize the influence of differences in the spatial distribution of the noise in parallel imaging acquisition. Both SNR and CNR equations are derived from the publicly available quality assurance pipeline (MRIQC) (Esteban et al., 2017).

Minor issues:

1. Line 49: This sentence is a phrase and not a complete sentence, it should be connected to the next sentence with a conjunction, in this case using not only…, but also… e.g., Scanning at a higher field strength not only holds the key to improving the intrinsic low SNR of ASL, but also recent research has...

Response: We have made the suggested changes accordingly.

Changes: “Scanning at a higher field strength might hold the key to improving the intrinsic low SNR of ASL as recent research has revealed that a higher static magnetic field (B0) leads to a supralinear improvement in SNR of a traditional MRI scan [10], which would also enhance the sensitivity and precision of ASL measurements.”

2. Line 341: The should be a semicolon before however instead of a comma, as it is being used as a conjunction.

Response: We have made the suggested changes accordingly.

Changes: “Additionally, we also evaluated the SNR across both ROIs and found a higher median for pTx compared to the 2Tx coil; however, …”.

3. Line 369: I think the flow of the paragraph will be improved if this sentence is start as.. Our fMRI experiment .. since you are previously describing other studies it can be a bit confusing when reading it at a first glance as The fMRI experiment…

Response: Thank you, we have made the suggested changes accordingly.

Changes: “Our fMRI experiment consistently elicited …”

 

Reviewer #3: Thank you for the corrections that you have made in this revision. The authors have mostly addressed my comments satisfactorily.

1. (R3.C5) There is still a concern on clarity with regards to the achieved B1+=15uT for the TR-FOCI RF pulses. The authors clarified that the pulse attributes were kept consistent across both coils. However, there is a discrepancy in the manuscript (line 144) which states, “For the 2Tx, a maximum B1+ of 12uT was used for the excitation pulses to accommodate amplifier limitations”. Given this amplifier limitation, how is the B1+=15uT achieved on the 2Tx?

Response: We thank the reviewer for the positive comments and suggestions. For the system used (7T Philips Achieva), when SAR limitations are encountered, changes in B1 maximum amplitude are applied only to the excitation pulses. Inversion pulse attributes, such as B1, pulse length, and flip angle are kept as predefined. Of course, the inversion pulses do not exceed safety limits. Please also check response R1.C1. We have added this notion to the manuscript.

Changes: For the pTx system, a maximum B1+ value of 18 μT was used for the excitation pulses; this is the default maximum for this system. For the 2Tx, a maximum B1+ of 12 μT was used for the excitation pulses to accommodate amplifier limitations. For the system used in the present study, changes in the B1 maximum are applied only to the excitation pulses. The inversion pulse attributes, such as B1 amplitude, pulse length, and flip angle were kept as predefined. The inversion pulses do not exceed safety limits. Apart from the excitation B1+ value, all sequence parameters were kept the same for all participants.

2. (R3.C6) Is it possible to show multiple slices of the B1 maps (S Fig. 1) which covers the labeling regions highlighted in Figure 1 to verify the claims of potential differences of the B1+ profile of the coil at the level of the tag?

Response: We have updated the Supplementary Figure 1 (S Fig. 1). 

Changes:

S Fig 1: Representative B1+ maps (DREAM B1) from an individual participant. Panel A depicts the B1+ maps in sagittal, coronal and axial planes. Panel B shows the location of four slices, corresponding to the typical location of the labeling plane. Noticeably higher B1 values are observed in the pTx compared to the 2Tx B1 map. Panel C displays the location of the four slices.

Finally, there are some minor inconsistencies with abbreviations (e.g. B1+ vs B1) used throughout the manuscript. Otherwise, I’m quite satisfied with the authors’ response.

Response: We thank the reviewer for pointing this out. We have made the suggested adjustments in the manuscript.

 

References

Dietrich, O., Raya, J.G., Reeder, S.B., Reiser, M.F., Schoenberg, S.O., 2007. Measurement of signal-to-noise ratios in MR images: Influence of multichannel coils, parallel imaging, and reconstruction filters. J. Magn. Reson. Imaging 26, 375–385. https://doi.org/10.1002/jmri.20969

Esteban, O., Birman, D., Schaer, M., Koyejo, O.O., Poldrack, R.A., Gorgolewski, K.J., 2017. MRIQC: Advancing the automatic prediction of image quality in MRI from unseen sites. PLOS ONE 12, e0184661. https://doi.org/10.1371/journal.pone.0184661

---

## [Editor Report · Decision Letter 2]

8 Aug 2024

Advancing 7T perfusion imaging by pulsed arterial spin labeling: Using a parallel transmit coil for enhanced labeling robustness and temporal SNR.

PONE-D-23-37867R2

Dear Dr. Oliveira,

We’re pleased to inform you that your manuscript has been judged scientifically suitable for publication and will be formally accepted for publication once it meets all outstanding technical requirements.

Kind regards,

Federico Giove, PhD

Academic Editor

PLOS ONE
---

## [Editor Report · Acceptance letter]

15 Aug 2024

PONE-D-23-37867R2 

PLOS ONE

Dear Dr. Oliveira, 

I'm pleased to inform you that your manuscript has been deemed suitable for publication in PLOS ONE. Congratulations! Your manuscript is now being handed over to our production team.

Kind regards, 

on behalf of

Dr. Federico Giove 

Academic Editor

PLOS ONE